# Coordinate Induction of Humoral and Spike Specific T-Cell Response in a Cohort of Italian Health Care Workers Receiving BNT162b2 mRNA Vaccine

**DOI:** 10.3390/microorganisms9061315

**Published:** 2021-06-16

**Authors:** Chiara Agrati, Concetta Castilletti, Delia Goletti, Silvia Meschi, Alessandra Sacchi, Giulia Matusali, Veronica Bordoni, Linda Petrone, Daniele Lapa, Stefania Notari, Valentina Vanini, Francesca Colavita, Alessandra Aiello, Alessandro Agresta, Chiara Farroni, Germana Grassi, Sara Leone, Francesco Vaia, Maria Rosaria Capobianchi, Giuseppe Ippolito, Vincenzo Puro

**Affiliations:** National Institute for Infectious Diseases (INMI) L. Spallanzani—IRCCS, Via Portuense 292, 00149 Roma, Italy; chiara.agrati@inmi.it (C.A.); concetta.castilletti@inmi.it (C.C.); delia.goletti@inmi.it (D.G.); silvia.meschi@inmi.it (S.M.); alessandra.sacchi@inmi.it (A.S.); giulia.matusali@inmi.it (G.M.); veronica.bordoni@inmi.it (V.B.); linda.petrone@inmi.it (L.P.); daniele.lapa@inmi.it (D.L.); stefania.notari@inmi.it (S.N.); valentina.vanini@inmi.it (V.V.); francesca.colavita@inmi.it (F.C.); alessandra.aiello@inmi.it (A.A.); alessandro.agresta@inmi.it (A.A.); chiara.farroni@inmi.it (C.F.); germana.grassi@inmi.it (G.G.); sara.leone@inmi.it (S.L.); francesco.vaia@inmi.it (F.V.); giuseppe.ippolito@inmi.it (G.I.); vincenzo.puro@inmi.it (V.P.)

**Keywords:** SARS-CoV2, mRNA vaccine, health care workers, whole blood T cell assay, coordinate immunity

## Abstract

Vaccination is the main public health measure to reduce SARS-CoV-2 transmission and hospitalization, and a massive worldwide scientific effort resulted in the rapid development of effective vaccines. This work aimed to define the dynamics of humoral and cell-mediated immune response in a cohort of health care workers (HCWs) who received a two-dose BNT162b2-mRNA vaccination. The serological response was evaluated by quantifying the anti-RBD and neutralizing antibodies. The cell-mediated response was performed by a whole blood test quantifying Th1 cytokines (IFN-γ, TNF-α, IL-2), produced in response to spike peptides. The BNT162b2-mRNA vaccine induced both humoral and cell-mediated immune responses against spike peptides in virtually all HCWs without previous SARS-CoV-2 infection, with a moderate inverse relation with age in the anti-RBD response. Spike-specific T cells produced several Th1 cytokines (IFN-γ, TNF-α, and IL-2), which correlated with the specific-serological response. Overall, our study describes the ability of the BNT162b2 mRNA vaccine to elicit a coordinated neutralizing humoral and spike-specific T cell response in HCWs. Assessing the dynamics of these parameters by an easy immune monitoring protocol can allow for the evaluation of the persistence of the vaccine response in order to define the optimal vaccination strategy.

## 1. Introduction

Large-scale vaccination is the single most effective public health measure for the mitigation of the coronavirus disease (COVID-19) pandemic. The vaccination strategy for the BNT162b2 vaccine involves a two-dose vaccination regimen administered 21 days apart, which has been demonstrated to induce a spike protein (S)-specific humoral and cellular immunity associated with a 95% efficacy in naïve individuals [1]. These data are confirmed in several reports focusing on health care workers (HCWs, [2,3]). Humoral immune response is considered the main immune correlate of protection, but the coordination between humoral- and cell-mediated arms has been clearly demonstrated to be more effective in fighting SARS-CoV-2 infection [4,5].

In this study, we addressed the coordinated humoral- and cell-mediated immune response to vaccination by assessing both arms of immunity in a cohort of HCWs who received a two-dose BNT162b2-mRNA vaccination at the National Institute for Infectious Diseases L. Spallanzani in Rome. The study was based on a two-step approach: in the first step, a small group of HCWs was longitudinally analyzed to establish the dynamics of immune response; in the second step, a cross-sectional evaluation was performed after the second dose to analyze the correlation between the two arms of immune response.

## 2. Materials and Methods

### 2.1. Enrolled Subjects

Two separate groups of HCWs who received the BNT162b2- mRNA vaccine were extracted from the cohort of vaccinated HCWs established at the INMI L. Spallanzani. The first group (longitudinal study) was a convenience sample of 35 HCWs tested before the first dose of vaccination (T0), before the second dose (T1), and 2 weeks after T1 (T2). The second, wider group (cross-sectional study) included 167 HCWs tested at T2. All enrolled HCWs were naïve for SARS-CoV-2 infection as shown by a negative test for anti-nucleocapsid (anti-N) and anti-spike receptor-binding-domain (anti-RBD) antibodies at T0. Out of 167 HCWs enrolled in the cross-sectional study, 119 (71%) were women, and the median age was 42 years (IQR 31–52). Most of the enrolled HCWs (*n* = 143, 86%) had been employed in the direct care of COVID-19 patients. The study was approved by the INMI Ethical committees (issue N. 297/2021), and all HCWs signed an informed consent.

### 2.2. Antibody Evaluation

Two commercial chemiluminescence microparticle antibody assays (ARCHITECT^®^ i2000sr Abbott Diagnostics, Chicago, IL, USA) were used: the anti-nucleoprotein IgG and the SARS-CoV-2 IgG II kit, which detected antibodies against the RBD of SARS-CoV-2.

### 2.3. Micro-Neutralization Assay (MNA_90_)

The assay was performed according to [6], using SARS-CoV-2/Human/ITA/PAVIA10734/2020, provided by Fausto Baldanti, Pavia, as a challenging virus. First, heat-inactivated and 7 two-fold serial diluted sera (starting dilution 1:10) were mixed and incubated at 37 °C 5% CO_2_ for 30 min with equal volumes of 100 TCID_50_ SARS-CoV-2. Next, 96-well tissue culture plates with sub-confluent Vero E6 cell monolayers were infected with 100 μL/well of virus-serum mixture and incubated at 37 °C and 5% CO_2_. To standardize the inter-assay procedures, positive control samples showing high (1:160) and low (1:40) neutralizing activity were included in each MNA session. After 48 h, microplates were observed using a light microscope for the presence of the cytopathic effect (CPE). The highest serum dilution inhibiting at least 90% of the CPE was indicated as the neutralization titer and was expressed as the reciprocal of serum dilution (MNA_90_).

### 2.4. T Cell Immune Response

Peripheral blood was collected in heparin tubes and stimulated with a pool of peptides spanning the spike protein (Miltenyi Biotech, Bergisch Gladbach, Germany) at 37 °C (5% CO_2_), according to [7]. A superantigen (SEB) was used as a positive control. Cultured plasma was harvested after 16–20 h of stimulation and stored at −80 °C. Th1-cytokines (IFN-γ, TNF-α, IL-2) were quantified in the plasma using an automatic ELISA (ELLA, Protein Simple). The detection limit of these assays were 0.17 pg/mL, 0.3 pg/mL, and 0.54 pg/mL for IFN-γ, TNF-α, and IL-2, respectively.

### 2.5. Statistical Analysis

Continuous variables including anti-RBD, anti-N, MNA_90_ titers, IFN-γ, TNF-α, and IL-2 levels were reported as median and interquartile range (IQR). The comparisons of the medians across groups were evaluated using Kruskal–Wallis analysis with the Mann–Whitney U-test with Bonferroni correction for pairwise comparisons. Correlations between the assays were assessed by non-parametric Spearman’s rank tests. To identify significant variables that could contribute towards the anti-RBD, MNA_90_, and IFN-γ response, a multiple linear regression model with a stepwise selection procedure was used. Analyses were performed in R. A 2-sided *p* value < 0.05 was considered statistically significant.

## 3. Results

We first assessed the kinetics of humoral- and cell-mediated immune responses to the BNT162b2-mRNA vaccination in 35 longitudinally sampled HCWs. Humoral response was evaluated by the anti-RBD antibody, while the natural infection was excluded by the anti-N antibody. As shown in Figure 1, the anti-N antibodies were undetectable at all time points, confirming no SARS-CoV-2 natural infection during the study duration (Figure 1a). In contrast, 100% of the HCWs presented detectable anti-RBD antibody response after both the first (T1) and second (T2) dose. Specifically, vaccination induced a significant increase of anti-RBD antibodies at T1 that further increased at T2 (median T0: 3.4 AU/mL (IQR: 2.2–9.050) vs. T1: 820 AU/mL (488.7–1570) vs. T2: 16,665 AU/mL (8739–32,702), *p* < 0.0001)). For the analysis of S-specific T cell response, Th1 cytokines (IFN-γ, TNF-α, IL-2) released after in vitro stimulation were measured (Figure 1b). Before vaccination, the vast majority (95%) of HCWs did not produce IFN-γ after S stimulation, confirming no previous exposure to SARS-CoV2. Only 3 HCWs showed a low IFN-γ (97 pg/mL, 51 pg/mL, 32 pg/mL) release at T0, paralleled by low TNF-α-release (22.0 pg/mL, 91.0 pg/mL, 16.7 pg/mL). A significant increase of all S-induced cytokines was observed in all HCWs after vaccination. In particular, IFN-γ was significantly increased at T1 and was further increased at T2 [median T0: 1.85 pg/mL (0.1–10.7) vs. T1: 116.2 pg/mL (61.1–196.4) vs. T2: 429.8 pg/mL (190.5–713.6), *p* < 0.0001] (Figure 1b). A similar trend was observed for IL-2 [median T0: 4.8 pg/mL (2.175–7.875) vs. T1: 209.7 pg/mL (98.15–510) vs. T2: 278.9 pg/mL (162–943), *p* < 0.0001].Additionally, TNF-α production peaked at T1 and plateaued thereafter (median T0: 0.1 pg/mL (0.1–14.68)) vs. T1: 53.85 pg/mL (19.05–129.7) vs. T2: 85.5 pg/mL (49.75–148.5), *p* < 0.0001).

We then focused on the correlations between the two arms of immune response and on possible determinants of optimal response, using a wide group of vaccine recipients (*n =* 167), cross-sectionally sampled at T2. In this larger group, a strong induction of anti-RBD Abs was confirmed in 100% of recipients (Figure 2a). Moreover, MNA_90_, performed on a subgroup of these HCW (*n =* 73), showed detectable neutralizing antibodies in all but one recipient (98.6%) (median: 80 reciprocal of dilution (IQR: 40–280)). Interestingly, the neutralization titer strongly correlated with the anti-RBD titers (r = 0.8103, *p* < 0.0001), suggesting a strict relationship between anti-RBD antibodies and their neutralization capabilities in BNT162b2-mRNA vaccinated HCWs (Figure 2a,b). Accordingly, cytokine producing S-specific T cells were induced after vaccination (Figure 2c), and the release of IFN-γ correlated with both TNF-α (r = 0.3724, *p* < 0.0001) and IL-2 (r = 0.5168, *p* < 0.0001; Figure 2d), suggesting a well-orchestrated S-specific Th1 response. Moreover, although weak, a positive correlation was also observed between IFN-γ and TNF-α levels with the anti-RBD titers (r = 0.2085, *p* = 0.0072 and r = 0.2034, *p* = 0.0088, respectively. Figure 2e) as well as between the TNF-α and MNA_90_ titers (r = 0.2344, *p* = 0.0459, Figure 2f).

To identify possible factors associated with the strength of anti-RBD, IFN-γ, and MNA_90_ immune responses, the impact of age, gender, and provision of direct care to COVID-19 patients was analyzed in the cross-sectional study results. No significant association was observed, except a moderate inverse relation with age in the anti-RBD response (anti-RBD <50y: 17,399.7 (11,240.4–26,391.4) AU/mL vs. > 50y: 14,021.7 (8119.1–22,795.1) AU/mL, *p* = 0.046). The multiple linear regression model did not modify these findings.

## 4. Discussion

In this study, we showed the ability of the BNT162b2-mRNA vaccine to induce a coordinate humoral and T cell-mediated immune response against spike peptides in virtually all healthy, young, and middle-aged adults, predominantly female, without previous SARS-CoV-2 infection, with only a moderate inverse impact of age >50 years.

In natural infection, the induction of a coordinate adaptive immune response, characterized by both neutralizing antibody production and T cell response, is effective in fighting the infection and represents a main feature of mild disease [4,8]. Moreover, T cells seem to persist longer than antibodies after recovery and therefore, can orchestrate long lasting memory immunity [9], as known for other coronavirus infections [10]. Integrated immune response represents the goal of vaccination strategy: to induce protective immunity closely analogous to the coordinated adaptive antiviral immune response seen in recovered patients after natural SARS-CoV-2 infection. In this context, Yellow Fever and smallpox vaccines represent good examples of inducing remarkably effective and long-lived immune protection through the induction of neutralizing antibodies and T cells with broad specificity, high magnitude, polyfunctionality, high proliferative potential, and long-term persistence [11].

The assays used in this study are easy and highly reproducible, and therefore, are compatible with the routine monitoring of vaccinated people. Indeed, T cell response was detected by a whole blood assay, the platform of which is similar to what is currently used to test T cell-specific responses against mycobacterium tuberculosis.

The definition of a standardized immune assay able to define the response to a vaccine can represent a key tool to design a standard routine monitoring protocol for vaccinated subjects (both healthy and fragile persons). Using these experimental tools, we showed that T cell response after vaccination is characterized by a coordinated production of all Th1 cytokines, with IFN-γ correlating with both TNF-α and IL-2. These data suggest the induction of T cells able to produce several Th1 cytokines that can mirror an effective response able to exert effector functions (e.g., IFN-γ, TNF-α) and to sustain the homeostasis of specific T cells (e.g., IL-2). Further studies on T cell-specific immunity induced by vaccination should be performed to evaluate their polyfunctionality. Indeed, a polyfunctional T cell response represents the most effective differentiation profile of specific T cells and has been described in convalescent COVID-19 patients associated with cytokine production and a stem-like memory phenotype [12]. IFN-γ and TNF-α production characterize Th1 specific cells, and IL-2 is essential for the homeostatic maintenance of a functional specific T cell response.

The low baseline production of all IFN-γ and TNF-α observed in 3 HCWs, possibly due to low exposure without seroconversion or to cross-reactivity, did not prevent optimal response after vaccination.

Overall, our study describes the ability of the BNT162b2 mRNA vaccine to elicit a coordinated neutralizing humoral and spike-specific T cell response in HCWs. Assessing the dynamics of these parameters by easy immune assays can represent a powerful tool to identify low/non-responder patients, especially in critical clinical settings, and to define the durability of the immune response in order to define optimal vaccination strategies.

## Figures and Tables

**Figure 1 microorganisms-09-01315-f001:**
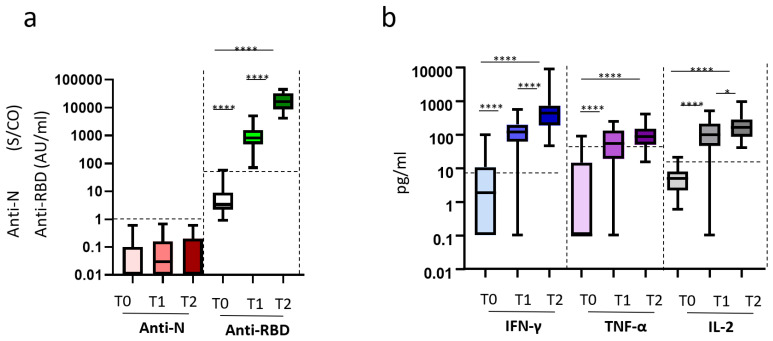
Kinetic of humoral and cell-mediated immune response to vaccination. Legend to Figure 1: Longitudinal study (**a**,**b**). Serum and peripheral blood were collected before (T0) vaccination, 3 weeks after the 1st dose (T1), and 2 weeks after the 2nd dose (T2). (**a**) SARS-CoV-2 specific anti-N Abs and anti-RBD Abs were quantified in sera samples at each time point and showed as a box and whiskers graph. Anti-N-IgG are expressed as index values S/CO, and values ≥ 1.4 are considered positive; Anti-RBD-IgG are expressed as arbitrary units AU/mL and values ≥ 50 are considered positive. (**b**) Cytokines (IFN-γ, TNF-α, IL-2) were quantified in stimulated blood samples at each time point and shown as median after subtracting the background. Dashed lines identify the cut-off of each test calculated as the mean +/− 2SEM of 55 anti-S and anti-N negative HCWs before vaccination. Results are shown as box and whiskers graphs. ****: *p* < 0.0001; *: *p* < 0.5.

**Figure 2 microorganisms-09-01315-f002:**
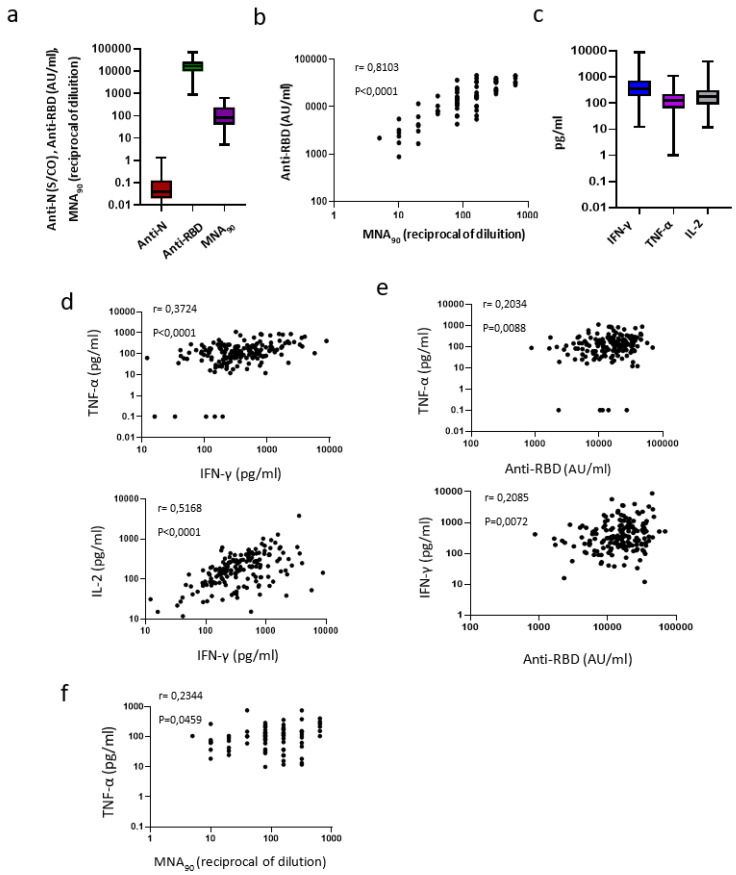
Coordinated immune response to vaccination. Legend to Figure 2: Cross-sectional study (**a**–**f**). (**a**) SARS-CoV-2 specific anti-N and anti-RBD Abs were analyzed at T2 in 169 HCWs; a microneutralization assay (MNA_90_) was also performed in a subgroup of the enrolled HVWs (*n =* 73). Results are shown as box and whiskers graphs. (**b**) The correlation between the anti-RBD antibody titer and MNA_90_ is shown (*n =* 73); (**c**) Cytokines (IFN-γ, TNF-α, IL-2) were quantified in stimulated blood samples at T2 (*n =* 169); (**d**) The correlation between the levels of IFN-γ and TNF-α and between IFN-γ and IL-2 are shown (*n =* 169). Each black dot represents one sample (**e**) The correlations between the level of IFN-γ and anti-RBD antibody titers and between the level of TNF-α and anti-RBD antibody titers are shown (*n =* 169). Each black dot represents one sample (**f**) The correlation between TNF-α and MNA_90_ is shown (*n =* 73). Each black dot represents one sample.

## Data Availability

The data presented in this study are available upon request from the corresponding author.

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
