# Peer review of "Coordinate Induction of Humoral and Spike Specific T-Cell Response in a Cohort of Italian Health Care Workers Receiving BNT162b2 mRNA Vaccine"

_microorganisms, 2021, doi:10.3390/microorganisms9061315_

Round 1

Reviewer 1 Report

This short study reports the results of immunization with the BNT162b mRNA vaccine in health care workers.  The study was performed well and the manuscript well written. However, the study appears to lack novelty and the authors should highlight how this study differs to other studies which report the results of vaccination with this vaccine.

The additional following points should be addressed;

All comments related to the induction  of multifunctional T cells should be removed because the correlations shown in figure 2 are generally weak and no direct measurement of polyfunctionality was performed.

The antibody data should be presented in antibody titers to allow comparison with previous studies and with the antibody titers elicited after infection with SARS-CoV-2.

Author Response

Reviewer 1

This short study reports the results of immunization with the BNT162b mRNA vaccine in health care workers.  The study was performed well and the manuscript well written. However, the study appears to lack novelty and the authors should highlight how this study differs to other studies which report the results of vaccination with this vaccine.

Answer. Thank you for the comments. We agree with the reviewer that several other studies have been recently published about the immune response to mRNA vaccine in health care workers (HCWs). Nevertheless, this paper shows some novelties about the immunological approach aimed to simplify the immune monitoring protocol and about the analysis of different Th1 cytokines produced after specific stimulation.

The majority of published papers focalized on the antibody response to BNT162b vaccine, while other studies looked also to the T cell response using the Elispot or Flow cytometry assays that need highly skilled laboratories. Here, we showed results on t cell-mediated immune response obtained through an easy assay on whole blood that does not need lymphocyte isolation and that could therefore be simply applied as a standard routinely monitoring protocol to the general population (both healthy and fragile subjects). This approach allows identifying non/low responder subjects and evaluating the kinetic of the immune response (both humoral and cell-mediated). These results could be useful to estimate the immune persistence and to evaluate possible strategies aimed in sustaining an effective and coordinated response overtime. In the literature, there are immune-based tests based on the whole blood platform applied to other infectious diseases, as those caused by Cytomegalovirus (CMV) or M. tuberculosis. Indeed, this whole blood assay has been exploited as tests for latent tuberculosis infection (Goletti et al, Respirology, 2018) or to monitor congenital CMV (Capretti MG et al., Clin Infect Dis 2020).

Moreover, the Spike specific T cell response was depicted by quantifying several Th1 cytokines after specific stimulation. The production and the coordination of several Th1 cytokines can indeed mirror an effective response able to exert effector functions (e.g., IFN-g, TNF-a) and to sustain the homeostasis of specific T cells (e.g. IL-2).

We now  highlighted these points in the discussion.

COMMENT: All comments related to the induction of multifunctional T cells should be removed because the correlations shown in figure 2 are generally weak and no direct measurement of polyfunctionality was performed.

Answer: According to the referee’s comment, we removed the comments related to the multifunctional T cell response and modified the line 129.

COMMENT: The antibody data should be presented in antibody titers to allow comparison with previous studies and with the antibody titers elicited after infection with SARS-CoV-2.

Answer. Thank you for the comment. For each time points, we quantified the anti-RBD and anti-N antibodies titer as Arbitrary Unit (AU)/ml and showed them as median values and IQR. The neutralization capability was quantified as the reciprocal of dilution and the median value and IQR have been added in the result section (line 125).

Reviewer 2 Report

In their study of immune responses induced by the BNT162b2 COVID-vaccine, Agrati and et al. analyzed both antibody and T cell responses to the vaccine-encoded S protein. They note that the assays they described are very easy to conduct, making them suitable for wide-spread use in COVID vaccine studies. While this statement is undisputed, the results they generated lack any novelty and the study did not generate any new insights: COVID-vaccine induced responses have been reported in a variety of other studies already, using simple readouts like these authors, or much more sophisticated approaches, and in a variety of cohorts, including health care workers (the population targeted in this study) as the authors themselves pointed out. Therefore, the results of this study simply confirm those obtained by others (and not the other way around, as the authors claim in line 39). The authors also overstate the implications of their findings: by only testing supernatants of T cell cultures stimulated with S protein-derived peptides, it is impossible to conclude that polyfunctional T cells were induced (to make that claim, simultaneous cytokine secretion by individual T cells would need to be shown, eg. by intracellular cytokine staining and flow cytometric analysis). It is also inappropriate to talk about a “coordinated induction of humoral and cellular responses” since the authors only show coincidence, not a causal link between the two (much less a “coordination”). One of the studies they reference drew conclusions about coordination of antibody and T cell responses based on the presence of Tfh cells, but in the present study, no T cell phenotypes were determined.

One of the stated goals of the study was to “identify possible determinant of optimal response[s]”. To draw such conclusions, the cohort would have had to been significantly larger and, as the authors acknowledge, it was a targeted – and, thus, highly skewed - population that lacks the kind of diversity (age range, gender, health status) needed to address this question. The only conclusion they could draw (although no data were shown) was that age inversely correlated with the strength of the vaccine-induced immune response, an observation that had already been made in numerous other studies involving much broader and more diverse target populations. The induction of a Th1-biased response by both approved RNA- vaccines had already been established in preclinical models as well as various other already published clinical studies, and a correlation between Th1-associated cytokines is also not a new or unexpected finding. The abstract raises an expectation in the reader that the study will address the hot topic of durability of the vaccine-induced response but while the authors provide some discussion of the topic, the question was not addressed experimentally and no conclusions are provided, other than that the proposed methodology could be used in the future to address the question.

Additional points:

  • Materials and methods:
    • The description of the microneut-assay is insufficient. While the authors provide a reference for the assay setup, it’s unclear how the data in Fig 2 b were obtained (and what the datapoints in that graph represent).
    • T cell response: the term “plasma” should NOT be used for culture supernatants but should be reserved for blood plasma!!
    • The description of the statistical analysis is confusing – it appears that the authors combined a parametric and non-parametric test (Mann-Whitney and Kruskal-Wallis). It is also unclear how the p-values for the graphs in Fig 2 d and e were obtained – all of them suggest high significance, but of what if there is no correlation between parameters (eg., IFN and anti-RBD abs)?
  • The figure legend for Fig2 is missing descriptions (panel a and b) and is, overall, inadequate (not enough detail).
  • In the discussion (line 168), the authors argue that the correlation between anti-RBD and viral neutralization justifies using ab-titers as a surrogate of biological activity. However, they failed to acknowledge a large body of literature showing a lack of correlation between those two parameters in other studies (eg., many COVID patients having high anti-RBD antibody titers that have no neutralizing activity, or low neutralizing activity of anti-RBD antibodies in the elderly, or the dependence of this correlation on the nature of the COVID vaccine/adjuvant)
  • The manuscript would greatly benefit from proof-reading by a native English speaker – there are numerous grammatical errors (singular/plural errors, awkward phrasing and word choices, missing verbs).

Author Response

Reviewer 2

Comment 1: In their study of immune responses induced by the BNT162b2 COVID-vaccine, Agrati and et al. analyzed both antibody and T cell responses to the vaccine-encoded S protein. They note that the assays they described are very easy to conduct, making them suitable for wide-spread use in COVID vaccine studies. While this statement is undisputed, the results they generated lack any novelty and the study did not generate any new insights: COVID-vaccine induced responses have been reported in a variety of other studies already, using simple readouts like these authors, or much more sophisticated approaches, and in a variety of cohorts, including health care workers (the population targeted in this study) as the authors themselves pointed out. Therefore, the results of this study simply confirm those obtained by others (and not the other way around, as the authors claim in line 39). The authors also overstate the implications of their findings: by only testing supernatants of T cell cultures stimulated with S protein-derived peptides, it is impossible to conclude that polyfunctional T cells were induced (to make that claim, simultaneous cytokine secretion by individual T cells would need to be shown, eg. by intracellular cytokine staining and flow cytometric analysis). It is also inappropriate to talk about a “coordinated induction of humoral and cellular responses” since the authors only show coincidence, not a causal link between the two (much less a “coordination”). One of the studies they reference drew conclusions about coordination of antibody and T cell responses based on the presence of Tfh cells, but in the present study, no T cell phenotypes were determined.

Answer. Thank you for the comments.  We agree with the referee that several other studies were published on the immune response to BNT162b2 vaccination in the HCW setting. Nevertheless, this manuscript showed novelties about the immunological approach aimed to simplify the immune monitoring protocol and about the analysis of different Th1 cytokines produced after specific stimulation.

The majority of the published papers focalized on the antibody response to BNT162b vaccine, while other studies looked also to the T cell response using the Elispot or Flow cytometry assays that need highly skilled laboratories. Here, we showed data about cell-mediated immune response obtained through an easy assay on whole blood that does not need lymphocyte isolation and that could therefore be simply applied as a standard routinely monitoring protocol to the general population (both healthy and fragile subjects). This approach allows identifying non/low responder subjects and evaluating the kinetic of the immune response (both humoral and cell-mediated). These results could be useful to estimate the immune persistence and to evaluate possible strategies aimed in sustaining an effective and coordinated response overtime. Moreover, the Spike specific T-cell response was depicted by quantifying several Th1 cytokines after specific stimulation. The production and the coordination of several Th1 cytokines can indeed mirror an effective response able to exert effector functions (e.g., IFN-γ, TNF-a) and to sustain the homeostasis of specific T cells (e.g. IL-2).  We highlighted these points in the discussion.

Moreover, we also agree that the polyfunctionality of T cells can be demonstrated by intracellular staining and flow cytometry. Accordingly, we removed the comment about the polyfunctional T cells and added a sentence about the need to perform polyfunctional analysis.

We also agree with the reviewer that we did not demonstrated a causal link between the strength of the cell-mediated and the humoral responses. Nevertheless, the positive correlation between the Spike-specific humoral and T cell response represents a marker of a synergic antigen-specific immune response.  We highlighted these points in the discussion.

Comment: One of the stated goals of the study was to “identify possible determinant of optimal response[s]”. To draw such conclusions, the cohort would have had to been significantly larger and, as the authors acknowledge, it was a targeted – and, thus, highly skewed - population that lacks the kind of diversity (age range, gender, health status) needed to address this question. The only conclusion they could draw (although no data were shown) was that age inversely correlated with the strength of the vaccine-induced immune response, an observation that had already been made in numerous other studies involving much broader and more diverse target populations. The induction of a Th1-biased response by both approved RNA- vaccines had already been established in preclinical models as well as various other already published clinical studies, and a correlation between Th1-associated cytokines is also not a new or unexpected finding. The abstract raises an expectation in the reader that the study will address the hot topic of durability of the vaccine-induced response but while the authors provide some discussion of the topic, the question was not addressed experimentally and no conclusions are provided, other than that the proposed methodology could be used in the future to address the question.

 Answer. Thank you for the comments.  We agree with the referee that a much larger study should be designed to identify determinants of optimal response. Therefore, we removed this sentence from the end of the introduction. In the present cohort, we analyzed the impact of age, gender and provision of direct care to COVID-19 patients. No significant association was observed, except a moderate inverse relation with age in the anti-RBD response [anti-RBD <50y: 17399.7 (11240.4-26391.4) AU/ml vs > 50y: 14021.7 (8119.1-22795.1) AU/ml, p= 0.046]. These results are shown in lines 134-137.

We also agree that the coordinated production of several Th1 cytokines is not an unexpected finding. Nevertheless, this is the first evidence that this can be measured in whole blood providing a tool easily reproduceable in routine laboratories and that  a wide Th1 response correlating also with the antibody production may represent a good point to be highlighted. Accordingly, we modify the abstract and the discussion section.

Additional points:

  • COMMENT: The description of the microneut-assay is insufficient. While the authors provide a reference for the assay setup, it’s unclear how the data in Fig 2 b were obtained (and what the datapoints in that graph represent).
  • Thank you for the comments. We agree with the referee that the microneutralization description could be improved and we added a brief description of the method. The graph 2b represents the correlation between anti RBD IgG and the MNA90 titers analyzed by non-parametric Spearman's rank tests. Datapoints in the graph represent the point of intersection between each neutralizing titer and anti RBD arbitrary units/milliliter.

  • COMMENT T cell response: the term “plasma” should NOT be used for culture supernatants but should be reserved for blood plasma!!

Answer. Thank you for the comments.  We kindly ask the referee to consider that the “plasma” is the correct term. The test is based on cultured whole blood, and after an overnight incubation we harvest “plasma” of “cultured blood. Also in diagnostic settings using whole blood tests, as in tuberculosis, the researchers write aboul detection of immune factors in “plasma” (Nemes, AJRCCM, 2017) Therefore, we would keep this term because appropriate in this context.

.

  • COMMENT The description of the statistical analysis is confusing – it appears that the authors combined a parametric and non-parametric test (Mann-Whitney and Kruskal-Wallis). It is also unclear how the p-values for the graphs in Fig 2 d and e were obtained – all of them suggest high significance, but of what if there is no correlation between parameters (eg., IFN and anti-RBD abs)?

Answer. Thank you for the comments.  To evaluate the statistical significance of our data, two different non parametric tests were used. At first, the comparison of medians across the three groups were evaluated by Kruskal–Wallis analysis. Then, Mann–Whitney U-test with Bonferroni correction (another non parametric test) was used to define the significance for pairwise comparisons. The correlation analysis was performed by using the non-parametric Spearman's rank tests. The values of p and r were detailed in the figure. This test is a nonparametric measure of the strength and direction of association that exist between two variables. We agree with the referee that the correlation strength (Rho value) for Fig 2 e-f is not high (r<0.3). Therefore, we modified the text adding “weak correlation” (line 130).

  • COMMENT: The figure legend for Fig2 is missing descriptions (panel a and b) and is, overall, inadequate (not enough detail).

Answer. Thank you for the comments.  We improve the figure legend for Figure 2

  • COMMENT: In the discussion (line 168), the authors argue that the correlation between anti-RBD and viral neutralization justifies using ab-titers as a surrogate of biological activity. However, they failed to acknowledge a large body of literature showing a lack of correlation between those two parameters in other studies (eg., many COVID patients having high anti-RBD antibody titers that have no neutralizing activity, or low neutralizing activity of anti-RBD antibodies in the elderly, or the dependence of this correlation on the nature of the COVID vaccine/adjuvant)

Answer. Thank you for the comments. We agree with the referee the functional properties of antibody should be analyzed by the neutralization assay and that in convalescent COVID-19 patients the anti-Spike titer and the neutralization assay do not have a strong correlation. In contrast, the data presented in this paper and other our submitted observations showed a very good correlation between the anti-RBD assay (Abbott ARCHITECT® i2000sr) and the neutralization titer in vaccinated subjects. These data allow the opportunity to expand the antibody analysis to a wide population as the neutralization assay is time consuming and requires highly skilled personnel and BSL-3 facility.

We specified this point in the discussion (line 208).

  • COMMENT: The manuscript would greatly benefit from proof-reading by a native English speaker – there are numerous grammatical errors (singular/plural errors, awkward phrasing and word choices, missing verbs).

Answer. Thank you for the comments.  The manuscript has been revised by a native English mother tongue person. 

Round 2

Reviewer 1 Report

The authors addressed my major concerns about this manuscript and the over-interpretation of the data has been corrected. I only have a few minor points which should be addressed .

I wondered if the HCW who failed to develop neutralizing antibodies showed a low anti-RBD titer in the Elisa.  Irrespective of the answer, I think that this point might be included in the results section.

My other points are trivial;

Line 49- change "were" to "was"

Line 61-nucleocapsid

Line 125-recipient

Line 196-infections

Author Response

Reviewer 1

We thank the reviewer for his revisions which improved the manuscript.

COMMENT: I wondered if the HCW who failed to develop neutralizing antibodies showed a low anti-RBD titer in the Elisa.  Irrespective of the answer, I think that this point might be included in the results section.

Answer: We agree with the referees that this is a major point to be highlighted. With this aim, we performed a correlation analysis between the anti-RBD levels and the neutralization titer. Results, shown in Figure 1D, reported a strong correlation between these two assays (r=0.8103, p>0.0001), suggesting that the vaccine-induced anti-RBD and their neutralization capability were strictly associated. Accordingly, the HCW that responded to vaccine producing low level of anti-RBD showed also a low level of neutralization antibodies. We better highlighted this point in the result section (line 125-128).

COMMENT: My other points are trivial; Line 49- change "were" to "was"; Line 61-nucleocapsid; Line 125-recipient; Line 196-infections

Answer: We modified the text according to the reviewer suggestions.

Reviewer 2 Report

The authors have made a serious effort to address the reviewers’ comments and concerns. While they have fixed several issues identified by both reviewers, the response, however, further highlights the main problem the  manuscript suffers from: it is presented as a research study (and the title unequivocally suggests that this is a research study), but the authors’ response makes it clear that the main objective is to describe and promote a methodology and technical approach, so this is really meant to be a “methods paper”. As such, it would not be expected to present novel scientific insights (both reviewers pointed out the lack of novelty and lack of new insights into COVID-vaccine induced immune responses that were produced). But, as a methods paper, the manuscript would need to be significantly re-written/re-structured to focus on the methodology rather than the target population. It would also require a better description of what “vaccine failure” looks like (ie, what type of ab and T cell response would you get in a poor responder – eg., nursing home residents/frail elderly vaccine recipients who received the standard vaccine regimen) to establish cutoffs. A methods-paper needs to provide more guidance to the reader about how to interpret results they may get when applying the method to their study and for that purpose, the population used for this study was too homogenous (with no examples of failed/poor vaccination – the authors report that 100% of their volunteers have strong RBD Ab-responses – line 123; no vaccine breakthrough cases; only one type of vaccine). I completely agree with the authors that their results (Fig 2b) show a nice correlation between antibody titer and viral neutralization, but again, they can only draw this conclusion for this particular vaccine and the homogenous target population used for the study (this correlation may not be seen with a non RNA-based vaccine, or in other populations, especially the elderly or immunocompromised vaccine recipients). Thus, recommending the use of a simple RBD-ELISA as a (general, for any COVID vaccine and any target population) surrogate readout of antibody functionality (line 208) is not appropriate and not justified based on the study’s design.

Author Response

Reviewer 2

COMMENT: The authors have made a serious effort to address the reviewers’ comments and concerns. While they have fixed several issues identified by both reviewers, the response, however, further highlights the main problem the  manuscript suffers from: it is presented as a research study (and the title unequivocally suggests that this is a research study), but the authors’ response makes it clear that the main objective is to describe and promote a methodology and technical approach, so this is really meant to be a “methods paper”. As such, it would not be expected to present novel scientific insights (both reviewers pointed out the lack of novelty and lack of new insights into COVID-vaccine induced immune responses that were produced). But, as a methods paper, the manuscript would need to be significantly re-written/re-structured to focus on the methodology rather than the target population. It would also require a better description of what “vaccine failure” looks like (ie, what type of ab and T cell response would you get in a poor responder – eg., nursing home residents/frail elderly vaccine recipients who received the standard vaccine regimen) to establish cutoffs. A methods-paper needs to provide more guidance to the reader about how to interpret results they may get when applying the method to their study and for that purpose, the population used for this study was too homogenous (with no examples of failed/poor vaccination – the authors report that 100% of their volunteers have strong RBD Ab-responses – line 123; no vaccine breakthrough cases; only one type of vaccine). I completely agree with the authors that their results (Fig 2b) show a nice correlation between antibody titer and viral neutralization, but again, they can only draw this conclusion for this particular vaccine and the homogenous target population used for the study (this correlation may not be seen with a non RNA-based vaccine, or in other populations, especially the elderly or immunocompromised vaccine recipients). Thus, recommending the use of a simple RBD-ELISA as a (general, for any COVID vaccine and any target population) surrogate readout of antibody functionality (line 208) is not appropriate and not justified based on the study’s design.suggestions.

Answer: We thank the reviewer for your useful revision that improved the manuscript. We confirmed that the submitted paper is a research study aimed to assess the humoral and cell-mediated immune response in a cohort of vaccinated Health Care Workers and to evaluate the correlation between these two arms of the immune system. We decided to use a simple T cell assay on whole blood that can be easily applied to a large population and performed also in laboratories not specifically skilled for lymphocytes isolation. We completely agree with the referee that a “method paper” would need a completely different experimental approach, the enrollment of more heterogeneous population and the comparison among different vaccine platforms. Although interesting, this was not the focus of this work but may represent an excellent suggestion for a further work.

We agree with the referees that the correlation between anti-RBD and the neutralization titer needs some more details and that we can not generalize this conclusion. Accordingly, we modified the result section (line 128) and delete the sentence related to the use of anti-RBD as a surrogate of neutralization (lines 209-211).